# Decoherence of $V_B^-$ spin defects in monoisotopic hexagonal boron nitride

A. Haykal [1], R. Tanos[1], N. Minotto[1], A. Durand [1], F. Fabre[1], J. Li [2], J. H. Edgar [2], V. Ivády[3,4], A. Gali [5,6], T. Michel[1], A. Dréau[1], B. Gil [1], G. Cassabois[1] & V. Jacques [1✉]

Spin defects in hexagonal boron nitride (hBN) are promising quantum systems for the design of flexible two-dimensional quantum sensing platforms. Here we rely on hBN crystals isotopically enriched with either [10]B or [11]B to investigate the isotope-dependent properties of a spin defect featuring a broadband photoluminescence signal in the near infrared. By analyzing the hyperfine structure of the spin defect while changing the boron isotope, we first confirm that it corresponds to the negatively charged boron-vacancy center ($V_B^-$). We then show that its spin coherence properties are slightly improved in [10]B-enriched samples. This is supported by numerical simulations employing cluster correlation expansion methods, which reveal the importance of the hyperfine Fermi contact term for calculating the coherence time of point defects in hBN. Using cross-relaxation spectroscopy, we finally identify dark electron spin impurities as an additional source of decoherence. This work provides new insights into the properties of $V_B^-$ spin defects, which are valuable for the future development of hBN-based quantum sensing foils.

---

[1] Laboratoire Charles Coulomb, Université de Montpellier and CNRS, Montpellier, France. [2] Tim Taylor Department of Chemical Engineering, Kansas State University, Manhattan, KS, USA. [3] Max Planck Institute for the Physics of Complex Systems, Dresden, Germany. [4] Department of Physics, Linköping University, Linköping, Sweden. [5] Wigner Research Centre for Physics, Budapest, Hungary. [6] Department of Atomic Physics, Budapest University of Technology and Economics, Budapest, Hungary. ✉email: vincent.jacques@umontpellier.fr

Optically active point defects in semiconductors are currently attracting intense research attention owing to prospective applications in the growing field of quantum technologies[1]. These defects, that feature localized electronic states with energy levels deeply buried inside the bandgap of the host material, behave as artificial atoms with efficient optical transitions. Importantly, the electronic spin of the defect can offer a fertile quantum resource for the development of a wide variety of applications. Besides its use as a quantum bit for the realization of advanced quantum information protocols[2–4], the electronic spin state can also be exploited for quantum sensing purposes, by taking advantage of its extreme sensitivity to external perturbations such as magnetic and electric fields, pressure or temperature[5]. In this context, the nitrogen-vacancy (NV) defect in diamond is undoubtedly the most advanced solid-state quantum sensing platform, which has already found numerous applications in condensed matter physics and life sciences[6–8]. However, the search for alternative material platforms that could expand the range of quantum sensing functionalities offered by diamond remains very active worldwide[9]. Along this research line, a particular attention is currently being paid to layered van der Waals crystals because such materials promise the design of flexible, atomically thin quantum sensing foils[10–13].

Among the broad variety of layered van der Waals crystals, hexagonal boron nitride (hBN) is an ideal material for hosting optically active spin defects owing to its large bandgap (~6 eV)[14]. During the last years, many papers have reported the isolation of individual defects in hBN, showing very bright optical emission and perfect photostability, even at ambient conditions[15–19]. However, the detection of the electron spin state associated with these point defects has long remained a missing ingredient towards the development of quantum sensors. This gap has been recently filled with the demonstration of optically detected magnetic resonance on two different types of defects, which have been tentatively assigned to negatively charged boron vacancy centers[20,21] and carbon-related impurities[22–24]. Identifying the microscopic structure of optically active spin defects remains however a highly difficult task given the large number of potential candidates, which can involve extrinsic substitutional or interstitial impurities as well as complex assemblies of vacancies and antisite defects. Density functional theory is a powerful tool to reduce the number of possible atomic structures through ab initio calculations of defect properties such as formation energies, charge state transition levels, photoluminescence (PL) spectra or electron spin densities[25–30]. On the experimental side, valuable informations can be obtained by analysing the hyperfine structure of the spin defect[20,23], which results from its interaction with neighboring nuclear spin species.

In hBN, each lattice site is occupied by an atom with a non-zero nuclear spin, in contrast to other materials such as diamond or silicon carbide. Nitrogen naturally occurs as the $^{14}$N isotope with a nuclear spin $I = 1$ (99.6% natural abundance), while boron has two stable isotopes, $^{11}$B with a nuclear spin $I = 3/2$ (80% natural abundance) and $^{10}$B with a nuclear spin $I = 3$ (20% natural abundance). To date, all the studies of spin defects in hBN were performed on crystals with the natural content of boron isotopes[20–24], making the interpretation of hyperfine spectra a difficult task[30]. In this work, we circumvent this problem by employing hBN crystals containing a single boron isotope. Using such monoisotopic crystals, we focus our study on a spin defect featuring a broadband PL signal centered at a wavelength of 800 nm. The analysis of its hyperfine structure while changing the boron isotope confirms that this defect corresponds to the negatively charged boron-vacancy center ($V_B^-$). We then investigate the spin coherence properties of $V_B^-$ centers hosted in monoisotopic hBN crystals. We observe a slight increase of the

spin coherence time $T_2$ in $^{10}$B-enriched samples, which is supported by numerical simulations using cluster correlation expansion methods. Using cross-relaxation spectroscopy, we finally identify dark electron spin impurities as an additional source of decoherence for the $V_B^-$ center in hBN.

## Results

**Sample description**. Monoisotopic, millimeter-sized h$^{10}$BN and h$^{11}$BN crystals were synthesized through the metal flux growth method described in ref. [31], while using isotopically enriched boron powders with either $^{11}$B (99.4%) or $^{10}$B (99.2%). The resulting crystals were irradiated with thermal neutrons to produce point defects. Two different processes can lead to the creation of defects via neutron irradiation: either by damages produced by neutron scattering through the crystal or via neutron absorption leading to nuclear transmutation. The latter process strongly depends on the isotopic content of the hBN crystal. Indeed, the thermal neutron capture cross-sections of $^{11}$B (~0.005 barn) and $^{14}$N (~1.8 barn) are orders of magnitude smaller than that of $^{10}$B (~3890 barn)[32]. As a result, neutron irradiation of the h$^{10}$BN crystal likely creates boron vacancy-related defects through nuclear transmutation and introduces interstitial $^7$Li atoms resulting from the fission of the $^{10}$B isotope[32,33]. Conversely, the h$^{11}$BN crystal is almost transparent to neutrons and point defects are solely created by neutron scattering. To partially compensate for the isotope-dependent efficiency of defect creation by neutron irradiation, the h$^{11}$BN crystal was irradiated with a dose of ~$2.6 \times 10^{17}$ n cm$^{-2}$ and the h$^{10}$BN crystal with a smaller dose of ~$2.6 \times 10^{16}$ n cm$^{-2}$.

**Optical characterization**. The monoisotopic content of our neutron-irradiated hBN crystals was first verified through Raman scattering spectroscopy. Figure 1a shows the Raman spectra recorded from the h$^{10}$BN and h$^{11}$BN crystals around the energy of the intralayer $E_{2g}$ phonon mode that corresponds to the in-plane vibration of boron and nitrogen atoms[34]. The energy of this phonon mode is at 1358 cm$^{-1}$ for h$^{11}$BN and is shifted to 1394 cm$^{-1}$ for h$^{10}$BN, as expected from the isotopic mass difference[35,36]. The full-width at half maximum of the Raman line is around 4.5 cm$^{-1}$, attesting for the high crystalline quality of the hBN samples. This value is however slightly larger than that usually measured on similar crystals not subjected to neutron irradiation (~3 cm$^{-1}$)[35].

The PL properties were then analyzed with a scanning confocal microscope operating at ambient conditions. A laser excitation at 532 nm was focused onto the sample with a high numerical aperture microscope objective (NA = 0.95) mounted on a three-axis piezoelectric scanner. The PL signal was collected by the same objective, focused in a 50-μm-diameter pinhole and finally directed either to a spectrometer or to a silicon avalanche photodiode operating in the single-photon counting regime. A lateral PL raster scan of the h$^{10}$BN crystal is shown in Fig. 1b. It reveals a uniform PL signal, which is characterized by a broadband emission spectrum centered around $\lambda \approx 800$ nm [Fig. 1d]. It was recently proposed to assign this emission line to negatively charged boron-vacancy centers ($V_B^-$)[20,21,28]. A similar PL signal was observed on the h$^{11}$BN crystal [Fig. 1d], though with an intensity decreased by roughly one order of magnitude despite a larger neutron irradiation dose. This confirms that point defects are efficiently created through nuclear transmutation in the h$^{10}$BN crystal. Interestingly, a confocal PL scan recorded along the depth of the crystal shows that optically active defects are tightly localized at the sample surfaces [Fig. 1c]. Here the width of the hBN layer producing a PL signal is indeed limited by the longitudinal spatial resolution of the confocal

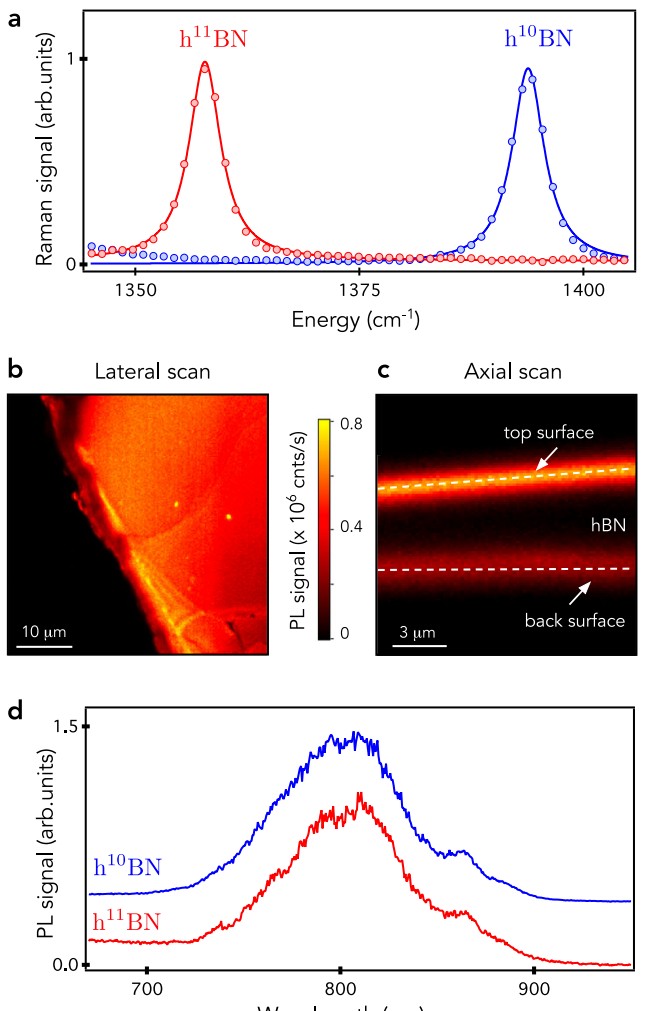

**Fig. 1 Optical properties of neutron-irradiated monoisotopic hBN crystals. a** Raman scattering spectra recorded at room temperature on the h$^{11}$BN (red dots) and h$^{10}$BN crystals (blue dots) after neutron irradiation. The solid lines are data fitting with Lorentzian functions. **b** Typical lateral PL raster scan of the h$^{10}$BN crystals under green laser illumination with a power of 1 mW. **c** Axial confocal scan showing a high PL signal localized at the crystal surfaces. **d** PL spectra recorded from the h$^{11}$BN (red) and h$^{10}$BN crystals (blue). The data have been normalized and vertically shifted for the sake of clarity.

microscope (~2 μm). This may be due either to the efficient migration of vacancy-related defects towards the surfaces or, more likely, to the stabilization of their charge state in an optically active configuration through a variation of the Fermi level close to the sample surface[37]. Although a precise understanding of this effect is beyond the scope of the present work, this is an important information for the future integration of exfoliated hBN layers doped with optically active spin defects into van der Waals heterostructures.

**Electron spin resonance spectroscopy.** The point defect associated with the emission line at $\lambda \approx 800$ nm exhibits very similar magneto-optical properties to those of the NV defect in diamond[20,21]. It features a spin triplet ground state $S = 1$, that can be polarized through optical pumping, coherently manipulated with a microwave excitation, and readout by recording its spin-dependent PL signal. These properties enable the detection of electron spin resonances (ESR) by optical means[20,21]. Optically

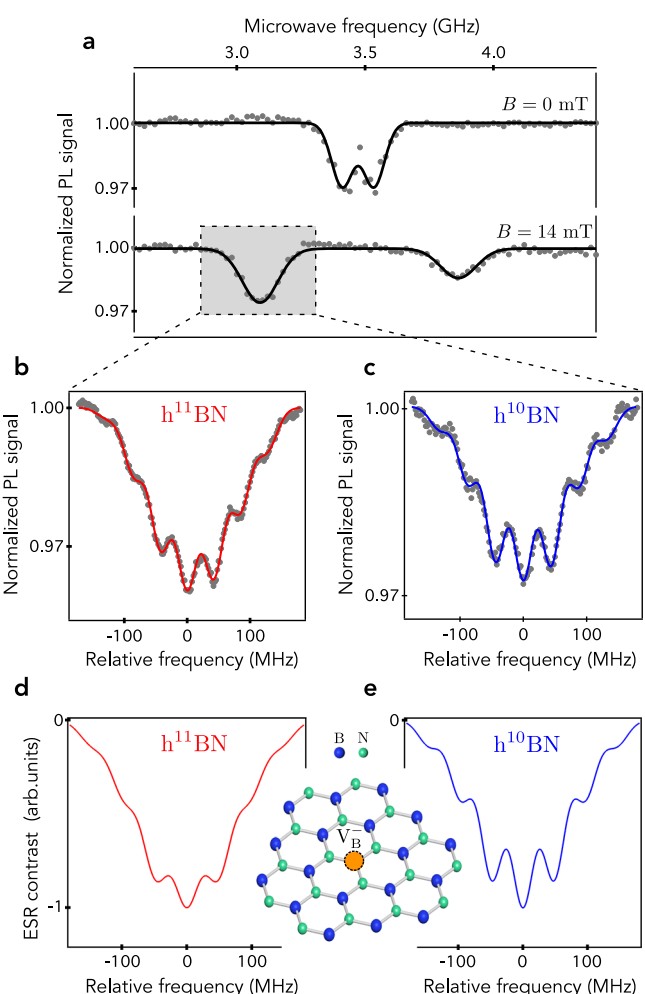

**Fig. 2 Analysis of the hyperfine structure. a** Optically detected ESR spectra recorded on the h$^{10}$BN crystal at zero field (top panel) and for a magnetic field $B = 14$ mT applied along the $c$ axis, i.e., perpendicular to the sample surface. **b, c** Hyperfine structure of the ESR line measured in **b** h$^{11}$BN and **c** h$^{10}$BN crystals. The solid lines are data fitting with a sum of seven Gaussian functions. **d, e** Simulated hyperfine spectra of the V$_B^-$ defect in hBN obtained via the procedure described in the main text while using $\Delta_s = 30$ MHz. The inset shows the atomic structure of the V$_B^-$ center in hBN.

detected ESR spectra were recorded by monitoring the PL signal while sweeping the frequency of a microwave field applied through a copper microwire directly spanned on the sample surface. When the microwave frequency crosses a transition between the ground state electron spin sublevels, the magnetic resonance is evidenced as a drop of the PL signal. At zero external magnetic field, the ESR frequencies are given by $\nu_\pm = D \pm E$, where $D$ and $E$ denote the longitudinal and transverse zero-field splitting parameters, respectively. A typical ESR spectrum recorded at zero field on the h$^{10}$BN crystal is displayed in Fig. 2a. From this experiment, we infer $D = 3.47$ GHz and $E = 61 \pm 1$ MHz, in good agreement with previous works[20,21].

**Analysis of the hyperfine structure.** To investigate the hyperfine structure of the spin defect, a static magnetic field was applied along the $c$ axis in order to shift the electron spin sublevels via the Zeeman effect and thus isolate a single ESR line in the spectrum [Fig. 2a]. The hyperfine structure recorded on the h$^{11}$BN crystal is displayed in Fig. 2b. Seven hyperfine lines can be resolved, with a

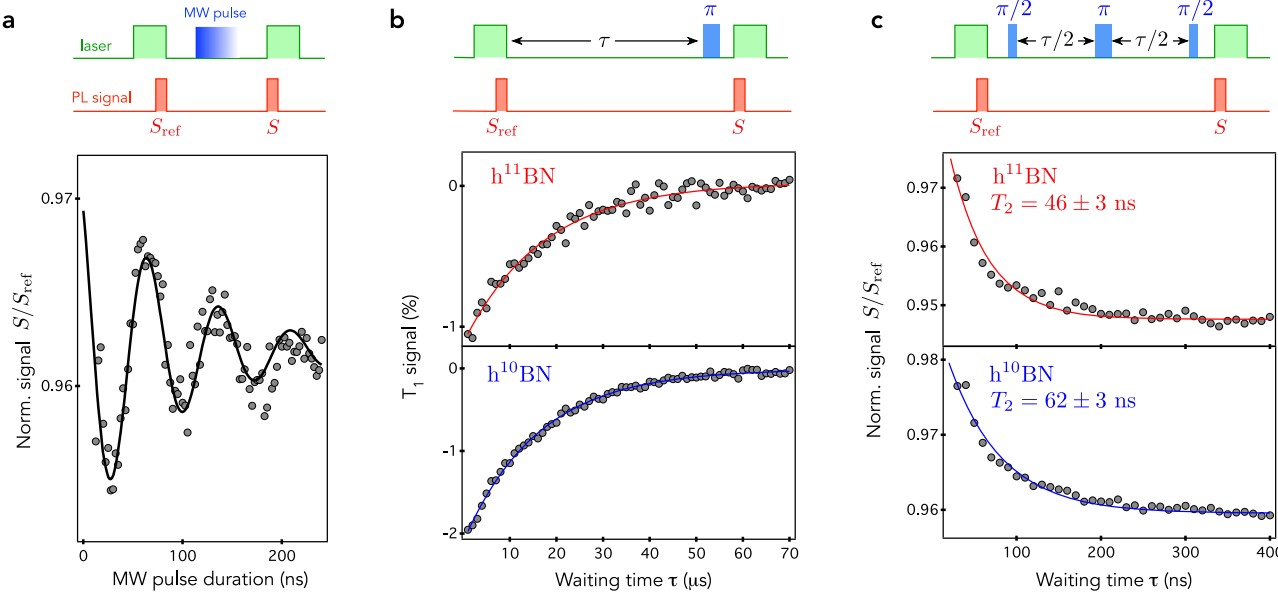

**Fig. 3 Spin coherence properties. a** Optically detected Rabi oscillations of an ensemble of $V_B^-$ spin defects recorded by applying the experimental sequence shown in the top panel. The spin-dependent PL signal $S$ is integrated at the beginning of the readout laser pulse and normalized by a reference signal $S_{ref}$, which corresponds to the steady-state PL signal measured at the end of the initialization laser pulse. **b** $T_1$ measurements in the two monoisotopic crystals. Here we plot the difference of the normalized PL signal $S/S_{ref}$ measured with and without applying the resonant microwave $\pi$-pulse (see top panel). Solid lines are data fitting with an exponential decay, leading to $T_1 \sim 16$ μs in both samples. **c** Measurement of the spin coherence time $T_2$ using a spin echo pulse sequence. Solid lines are data fitting with an exponential decay. All experiments are performed with a static magnetic field $B \sim 15$ mT applied along the $c$ axis. The duration of the laser pulses is 8 μs with a power of 1 mW and the integration time window of $S$ and $S_{ref}$ is fixed to 300 ns. The duration of the microwave $\pi/2$ pulse is set to 15 ns.

characteristic splitting $\mathcal{A} \sim 44$ MHz. This structure is analogous to that observed in hBN crystals with a natural content of boron isotopes, that corresponds to 80% of $^{11}$B[20,21]. The same experiment was performed for spin defects hosted in the monoisotopic h$^{10}$BN crystal, leading to a very similar hyperfine spectrum [Fig. 2c]. This observation unambiguously indicates that boron atoms are not involved in the seven-line structure of the hyperfine spectrum, and are therefore not localized at the first neighboring lattice sites of the spin defect. Considering the intrinsic nuclear spin species in hBN, the hyperfine structure can only be explained by the interaction of a central electronic spin with three equivalent $^{14}$N nucleus ($I = 1$). This situation can be obtained either for a boron-vacancy center or for a substitutional impurity with zero nuclear spin localized at a boron site. Substitutional carbon $C_B$ could be a plausible candidate, since carbon is a well-identified contaminant in hBN with a dominant spinless isotope ($^{12}$C, 99%). However $C_B$ defects exhibit a very different hyperfine structure, as discussed in ref. [30]. We thus conclude that the broad emission line centered at a wavelength of 800 nm in hBN corresponds to the $V_B^-$ center, as proposed in recent experimental works[20,21] and supported by ab initio theoretical calculations[28].

The hyperfine interaction of the $V_B^-$ center with the six equivalent boron atoms placed in the second neighboring lattice sites cannot be resolved and leads to a broadening of the ESR lines. At first glance, it might be surprising to observe similar linewidths in h$^{11}$BN and h$^{10}$BN crystals, because the $^{10}$B isotope has a higher nuclear spin ($I = 3$) than $^{11}$B ($I = 3/2$). This is due to the difference in nuclear gyromagnetic ratio between the two isotopes, $\gamma_n = 4.6$ MHz/T for $^{10}$B and $\gamma_n = 13.7$ MHz/T for $^{11}$B. Since the hyperfine coupling strength scales linearly with $\gamma_n$, it is reduced by a factor of three for $^{10}$B, which compensates for the impact of a higher nuclear spin in the broadening of the ESR line. To illustrate further this effect, simple simulations of the ESR spectra were performed. An exact calculation using a direct

diagonalisation of the spin Hamiltonian cannot be done with conventional computer ressources given the large number of nuclear spins involved. We therefore rely on a simplified model for which we only consider the longitudinal component $\mathcal{A}_{zz}$ of the hyperfine tensor. Here $z$ denotes the quantization axis of $V_B^-$, which corresponds to the $c$-axis of the hBN crystal[20]. The hyperfine coupling constants were taken from the ab initio calculations reported in ref. [28]. The simulation includes (i) the three equivalent $^{14}$N first-neighbors with $\mathcal{A}_{zz} = 48$ MHz, (ii) the six second-neighbors boron atoms with $\mathcal{A}_{zz} = -4.6$ MHz (resp. $-1.5$ MHz) for $^{11}$B (resp. $^{10}$B), and (iii) six equivalent $^{14}$N third-neighbors with $\mathcal{A}_{zz} = 4.5$ MHz. After calculating the position of the hyperfine lines, a convolution was applied with a Gaussian profile to take into account the inhomogeneous broadening of the $V_B^-$ electron spin transitions. The hyperfine spectra were simulated for various full-width at half maximum $\Delta_s$ of the Gaussian function. In Fig. 2d, e, we display the results obtained with $\Delta_s = 30$ MHz for which a fair agreement with experimental data is obtained. These simulations illustrate how the weaker hyperfine interaction with $^{10}$B leads to a similar width of the hyperfine lines despite a higher nuclear spin.

**Spin coherence properties.** We now study the spin coherence properties of an ensemble of $V_B^-$ defects hosted in monoisotopic hBN crystals. To this end, Rabi oscillations were first recorded by using the experimental sequence shown in Fig. 3a. After applying a first laser pulse to polarize the $V_B^-$ electron spins by optical pumping, coherent spin manipulation is performed by applying a resonant microwave pulse of varying duration, and the resulting spin state is finally readout optically by recording the spin-dependent PL signal. A typical Rabi oscillation of the $V_B^-$ electron spins is shown in Fig. 3a. This experiment provides a calibration of the duration of the microwave pulses ($\pi$ and $\pi/2$) employed in

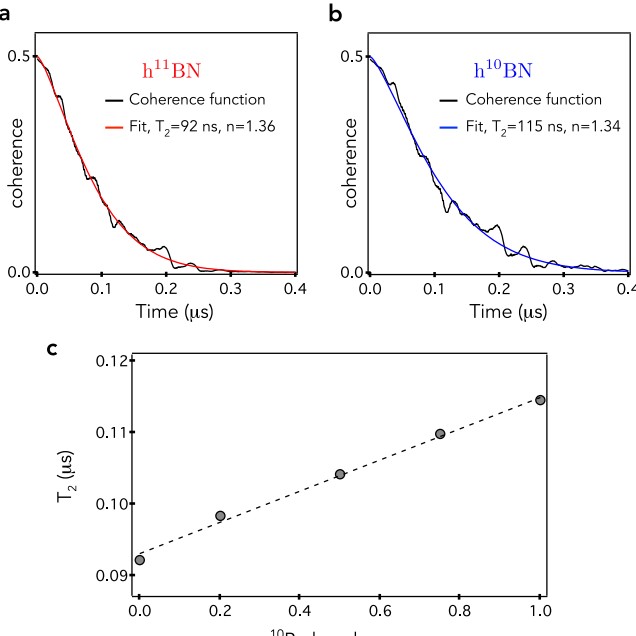

**Fig. 4 Nuclear spin bath induced decoherence. a, b** Simulated spin echo decay curve and corresponding stretched exponential fit for the $V_B^-$ defect in **a** h$^{11}$BN and **b** h$^{10}$BN. **c** Calculated spin coherence time $T_2$ as a function of the abundance of $^{10}$B isotope. The $T_2$ time is enhanced by the reduced hyperfine and nuclear spin-nuclear spin interaction strengths in $^{10}$B-enriched samples. The black dashed line is a linear fit.

pulsed ESR measurements. We note that the decay of the Rabi oscillation cannot be used to directly infer the electron spin dephasing time $T_2^\star$. Indeed, Rabi measurements mostly probe populations of the quantum system. As a result, the decay of the Rabi oscillation is usually much longer than $T_2^\star$ [38,39] (see Supplementary Fig. 1). The longitudinal spin relaxation time $T_1$ was measured by applying the usual experimental sequence sketched in Fig. 3b. Data fitting with an exponential decay leads to $T_1 \sim 16$ μs for the two monoisotopic hBN crystals. This value, which is consistent with that recently reported in the literature, is limited by the interaction of the $V_B^-$ electron spin with lattice phonons[21]. A spin echo sequence was then applied to infer the spin coherence time $T_2$ of the ensemble of $V_B^-$ centers [Fig. 3c]. By fitting the measured echo signal with an exponential decay, we obtain $T_2 = 46 \pm 3$ ns for the h$^{11}$BN crystal and $T_2 = 62 \pm 3$ ns for the h$^{10}$BN crystal.

**Theoretical analysis**. The coherence time of spin defects in solids is commonly limited by the magnetic noise produced by a bath of fluctuating paramagnetic impurities, which includes electron spin impurities in the host material, dangling bonds at the sample surface and nuclear spins. To better understand the origin of the short spin coherence time observed for $V_B^-$ centers in our monoisotopic hBN crystals, we first investigate the impact of the nuclear spin bath by performing numerical simulations with the generalized cluster correlation expansion method (gCCE)[40,41]. Within this theoretical framework, one can account for decoherence effects of different levels of complexity. When considering a bath dominated by nuclear spin species, the most common sources of dephasing are (i) nuclear spin flip-flops driven by the hyperfine interaction with the central electron spin, (ii) nuclear spin flip-flops induced by nuclear spin-nuclear spin dipolar interaction, and (iii) many-body correlation effects[40,41]. We find that all these effects up to three-nuclear spin correlation

contribute to the decoherence of the $V_B^-$ center in hBN. We therefore rely on the third order approximation (gCCE3) in our simulations (see Supplementary Note 1 and Supplementary Figs. 2 and 3). In addition, we use the hyperfine coupling constants calculated in ref. [28], which include the Fermi contact interaction. While this term is often neglected owing to the highly localized nature of the electron spin density associated with point defects in semiconductors[42], it plays an important role in the decoherence of the $V_B^-$ center in hBN, as discussed below.

The simulated spin echo decay curves of the $V_B^-$ center in h$^{11}$BN and h$^{10}$BN are shown in Fig. 4a, b. Both curves are fitted with a stretched exponential function, $\exp[-(t/T_2)^n]$, leading to $T_2 = 92$ ns and $T_2 = 115$ ns for h$^{11}$BN and h$^{10}$BN, respectively, with an exponent $n \sim 1.35$. The dependence of the coherence time with the isotopic content is depicted in Fig. 4c. It reveals a linear increase with the $^{10}$B abundance. This effect results from the reduced nuclear gyromagnetic ratio of $^{10}$B, that weakens the hyperfine interaction and the boron nuclear spin flip-flop rate, both of which has a positive impact on the coherence time of the central spin. The convergence of the simulation with respect to the size of the nuclear spin bath provides important additional informations on the decoherence processes. Indeed, our results indicate that the coherence time is mostly limited by the fluctuations of nuclear spins that are not farther than 0.8 nm from the central spin (see Supplementary Fig. 4). We thus conclude that decoherence of the $V_B^-$ electron spin in a nuclear spin bath is governed by highly localized interactions. As a result, nuclear spin flip-flops driven by the hyperfine interaction, which includes both the Fermi contact and the dipolar terms, play a critical role in the decoherence process. If the contact interaction and the first order coherence contribution are neglected in the simulation, we obtain a spin coherence time up to 23 μs, a value in line with earlier theoretical works[42]. The two orders of magnitude difference in the calculated coherence times, however, clearly falsifies the validity of these approximations. In fact, by taking the Fermi contact term into account, the $\mathcal{A}_{zz}$ hyperfine term of the first and second neighbor shell atoms of the $V_B^-$ center increase by a factor of ~28 and 2.7, respectively. Since the phase shift induced by the nuclear spin flip-flops is determined by the $\mathcal{A}_{zz}$ hyperfine terms, neglecting the Fermi contact has a drastic impact on the spin coherence function of all CCE orders. Furthermore, the first order CCE coherence function (C1) can solely be neglected when the Zeeman splitting of the nuclear spins exceeds the strength of the hyperfine coupling. Since the latter is in the 10 MHz range for neighboring nuclear spins, the required magnetic field to suppress the first order coherence function falls far beyond reach, in the 10–100 Tesla region. These results strikingly illustrates that the Fermi contact term and the first order coherence function cannot be neglected in the calculation of the coherence properties of spin defects in hBN.

**Cross-relaxation spectroscopy**. Even though our experimental results reproduce the slight increase of the spin coherence time in h$^{10}$BN, the measured $T_2$ values are about 50% smaller than the theoretical predictions [Fig. 3c]. This discrepancy could be explained by an additional source of decoherence in the experiments, such as electron spin impurities in the environment. Several paramagnetic centers with electronic spin $S = 1/2$ have been identified in neutron-irradiated hBN crystals through electron paramagnetic resonance (EPR) measurements[43–45]. Some of these defects were recently assigned to $N_BV_N$ and $C_NV_B$ centers[45]. To probe the presence of such $S = 1/2$ paramagnetic impurities in the vicinity of $V_B^-$ centers, we measured the longitudinal spin relaxation time $T_1$ as a function of a static magnetic field applied along the $c$ axis. When the ESR transition of the $V_B^-$ center is

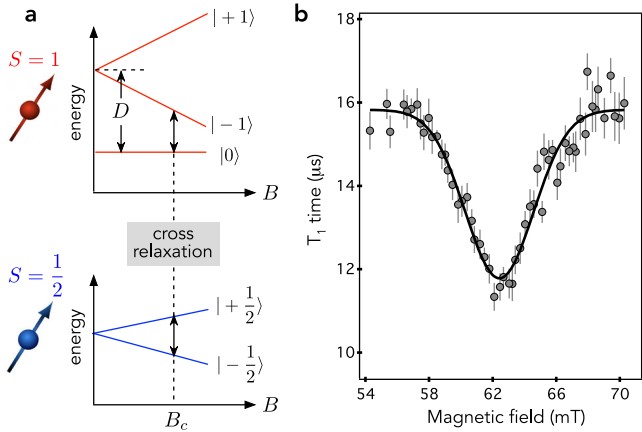

**Fig. 5 Cross-relaxation spectroscopy. a** Energy levels of the $V_B^-$ spin triplet ground state (top panel) and of a paramagnetic impurity with $S = 1/2$ (bottom panel) as a function of a static magnetic field $B$. The two spin systems are brought in resonance for $B = B_c \sim 62$ mT. **b** $T_1$ time of the $V_B^-$ spin defect in the $h^{10}$BN crystal as a function of a magnetic field applied along the $c$-axis. The field amplitude is inferred by recording the Zeeman shift of the ESR frequency of the $V_B^-$ defect. The solid line is data fitting with a Gaussian function. The error bars correspond to the uncertainty of fitting the $T_1$ decay curve with an exponential function.

brought in resonance with a target spin, cross-relaxation is induced by dipole-dipole interaction, which leads to a reduction of $T_1$ [Fig. 5a][46–48]. Considering a paramagnetic center with $S = 1/2$, the resonance condition is obtained for a magnetic field $B_c = D/2\gamma_e$, where $\gamma_e = 28$ GHz/T is the electron spin gyromagnetic ratio. For the $V_B^-$ center, it corresponds to $B_c \sim 62$ mT. The experimental results are displayed in Fig. 5b. As anticipated, a drop of the longitudinal spin relaxation time is observed around 62 mT, which reveals the existence of dark electron spin impurities with $S = 1/2$ in the close vicinity of the $V_B^-$ defects. The reduced value of the $T_2$ time in our experiments compared to theoretical predictions is attributed to fluctuations of these paramagnetic centers.

We note that the few studies found in the literature on the spin coherence properties of $V_B^-$ centers in hBN provide very different results. In ref. [49], a coherence time around $T_2 \sim 80$ ns was measured in a neutron-irradiated hBN samples with a natural content of boron isotopes. This value is analogous to that obtained in the present work and is supported by our theoretical predictions. On the other hand, refs. [21,50] have reported the observation of a few μs-long coherence time in similar hBN samples. In these studies, the contrast of the echo signal was however very weak and the experimental data at short time time scale were not shown. Such long $T_2$ values, which are not captured by theoretical simulations using CCE methods, could be alternatively attributed to nuclear spin coherence.

## Discussion

To conclude, we have illustrated how hBN crystals isotopically enriched with either $^{10}$B or $^{11}$B can be used to identify the structure of spin defects through the analysis of their hyperfine structure. Beyond the $V_B^-$ center studied in this work, these methods could be applied to other spin impurities in hBN, such as carbon-related defects whose microscopic structure remains unidentified[22–24,30]. An in-depth study of the spin coherence properties of $V_B^-$ centers indicates a slight increase of $T_2$ time in $^{10}$B-enriched crystals, which is supported by numerical simulations using cluster correlation expansion methods. These simulations highlight the importance of the spin density

distribution, the corresponding Fermi-contact hyperfine fields, and first order correlation effect for determining the coherence time of solid-state spin qubits interacting with a dense nuclear spin bath. This finding, which generally applies to any defect hosted in a solid with a high nuclear spin density, constrains the validity of prediction of spin coherence times using hypothetical defects[51]. Using cross-relaxation spectroscopy, we finally identified dark electron spin impurities as an additional source of decoherence in hBN. Although such paramagnetic impurities have a detrimental effect of the spin coherence time, they could also be turned into a useful resource for quantum sensing applications[52]. This work provides key insights into the spin relaxation dynamics of $V_B^-$ centers in hBN, which is a promising spin defect for the future development of flexible, atomically thin quantum sensing foils.

## Data availability

The datasets generated as part of the current study are available from the corresponding authors upon reasonable request.

## Code availability

The codes used for the analysis included in the current study are available from the corresponding authors upon reasonable request.

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

## Acknowledgements

This work was supported by the French Agence Nationale de la Recherche under the program ESR/EquipEx+ (grant number ANR-21-ESRE-0025), the Institute for Quantum Technologies in Occitanie through the project BONIQs, the European Union H2020 Quantum Flagship Program under Grant Agreement No. 820394 (ASTERIQs), and by the U.S. Department of Energy, Office of Nuclear Energy under DOE Idaho Operations Office Contract DE-AC07-051D14517 as part of a Nuclear Science User Facilities experiment. The crystal growth (J.L. and J.H.E.) in this study was supported by the Materials Engineering and Processing program of the National Science Foundation, Award Number CMMI 1538127. We acknowledge the support of The Ohio State University Nuclear Reactor Laboratory and the assistance of Susan M. White, Lei Raymond Cao, Andrew Kauffman, and Kevin Herminghuysen for the irradiation services provided. The simulations (A.G. and V.I.) have been performed using the resources provided by the Hungarian Governmental Information Technology Development Agency and by the Swedish National Infrastructure for Computing (SNIC) at the National Supercomputer Centre (NSC). V.I. acknowledges the support from the MTA Premium Postdoctoral Research Program and the Knut and Alice Wallenberg Foundation through WBSQD2 project (Grant No. 2018.0071). A.G. acknowledges the National Research, Development, and Innovation Office of Hungary Grant No. KKP129866 of the National Excellence Program of Quantum-Coherent Materials Project and the Quantum Information National Laboratory supported by the Ministry of Innovation and Technology of Hungary. Raman spectra were recorded at IR-RAMAN Technological Platform of Montpellier University.

## Author contributions

V.J. conceived and coordinated the project. A.H., R.T., N.M., A.Du., F.F., A.Dr., T.M., B.G, G.C., and V.J. conducted and analyzed the experiments. J.L. and J.H.E. performed the crystal growth. V.I. and A.G. performed the numerical simulations. V.I. and V.J. wrote the manuscript. All the authors discussed the data and commented the manuscript.

## Competing interests

The authors declare no competing interests.
