## [Peer Review File · Nature Communications]

REVIEWER COMMENTS

Reviewer #1 (Remarks to the Author):

The Authors report on a joint experimental-theoretical research effort to probe the decoherence of boron monovacancy defects in hBN. hBN has been the subject of intense research efforts to identify and utilize quantum defects embedded in the material. In this work, the Authors first utilize monoisotopic hBN to strengthen the attribution of the broad 800 nm optical signature that exhibits ODMR to the boron monovacancy. Their argument is based on the hyperfine coupling to the neighboring nitrogen spins. To my pleasure, the Authors also addressed the superhyperfine coupling, which supports their argument. The Authors then study the spin coherence properties of the boron monovacancy and extract two different T₂ times in the different monoisotopic crystals. They attribute the difference in the T₂ times to the differing isotopes and demonstrate the importance of including the Fermi contact term in the CCE calculation, correcting a misconception in the literature. Lastly, the Authors probe additional sources of coherence, “dark spins”, which introduce additional decoherence beyond the nuclear bath.

I find the work to be of high quality and fairly well done. A number of issues are addressed, and I was happy to see that the concerns I had while reading were mostly addressed soon after in the manuscript. I just have one concern that should be addressed prior to publication because it is essential to the message of the manuscript.

The Authors give the impression that the T₂ times are extracted from a single defect in each of the two different samples that differ in isotopic purity. The difference in these T₂ times is attributed to the differing isotopes. However, it is premature to make such a conclusion if only two defects have been investigated. Differing local environments could instead lead to the differing T₂ times. Indeed, the Authors point out that other experiments have measured a range of T₂ values. To make an attribution to the isotope effect, the Authors should investigate a statistically significant number of defects to rule out the differing local environments and isolate the isotope effect. If this was already done, the Authors need to make this clear.

Reviewer #2 (Remarks to the Author):

The authors of this work reported their creation and study of boron-vacancy spin defects in isotopically enriched hexagonal boron nitride (hBN). Spin defects in hBN provide a new paradigm of

quantum systems (in van der Waals materials) that will have applications in quantum sensing and other fields. Previous research has shown that isotopic enrichment can increase the coherence of diamond nitrogen-vacancy centers. Thus we would hope isotopic enrichment can increase the coherence time of spin defects in hBN or add new functionalities. In this work, the authors used hBN crystals enriched with either boron-10 or boron-11 isotopes to study isotope-dependent properties of spin defects. By comparing the hyperfine structures with different boron isotopes, the authors confirmed that the created spin defects are negatively-charged boron vacancies as claimed by other works before. However, the coherence time of hBN spin defects reported in this work is much shorter than several former works and does not show the advantages of isotopic enrichment.

The authors attributed the discrepancy between their result of short electron spin coherence time (T_2) and several former results of much longer coherence times to possible mistakes made by other groups. However, I found Fig. 3(c) was inconsistent with Fig. 3(a) in Figure 3 (Spin coherence properties) of this work. Fig. 3(a) in this work shows clear Rabi oscillations for times longer than 200 ns. Based on Fig. 3(a), the inhomogeneous coherence time (T_2^*) is about 100 ns, consistent with former results. T_2 must be longer than T_2^* . Thus the coherence time T_2 should be much longer than 100 ns. However, Fig. 3(c) shows the T_2 to be only about 46 ns or 62 ns, much shorter than the T_2^* shown in Fig. 3(a). The inconsistency between Fig. 3(a) and Fig. 3(c) may be because the fast decay shown in Fig. 3(c) is not due to spin decoherence. For example, the laser and MW pulses might have finite rise and decay times and overlaps that affect the measurement. In addition, to measure T_2 , the MW pulse durations should be much shorter than T_2 . If the MW pulse durations were not short enough, the spins might have significant decoherence within each pulse.

The paper “Defect Engineering of Monoisotopic Hexagonal Boron Nitride Crystals via Neutron Transmutation Doping” [Chem. Mater. 2021, 33, 9231–9239] has reported ODMR with isotopically enriched hBN and should be cited in this work.

In summary, I think some conclusions of this work are not supported by its results, especially the inconsistency between Fig. 3(a) and Fig. 3(c). Isotopic enrichment may become important for hBN spin defects in the future. However, the current work has not shown clear advantages of isotopic enrichment. Thus I cannot recommend the publication of this work in Nature Communications.

Reviewer #3 (Remarks to the Author):

This manuscript from A. Haykal et al. reports on the experimental measurements of the spin coherence in isotopically enriched hBN samples. This work studies how the coherence properties of the VB- defect in hBN change when the material contains ^{10}B or ^{11}B . The

authors interface experimental and theory to identify the spin structure of the defects through the analysis of the hyperfine structure. Overall, this paper is well written, the results are interesting and presented clearly. I think the topic is suitable for publication in Nature Communications. However, I have one major comment that should be addressed before I can recommend publication. The authors claim to unambiguously confirm that the investigated defects, created by thermal neutron irradiation, correspond to the VB⁻. This claim is based on the observation of 7 hyperfine lines in the ODMR spectrum. As the authors state at pag.8, this can be explained either by a B vacancy or a substitutional impurity with I=0. It seems that the conclusion about the observation of B vacancy is based on the fact that neutron transmutational doping with irradiation “likely produces boron vacancy-related centers”. Although the creation of B vacancies is likely, this does not seem to exclude the possibility to have also created substitutional defects with I=0. At pag. 15, the authors comment on how neutron irradiation can create other defects centers with S=1/2, including defects with C inclusion. I think this point needs to be clarified in the paper. In order to “unambiguously” claim the VB⁻ defects, I would suggest to discuss why no other substitutional defects with I=0 could be created during the neutron irradiation.

Authors' Rebuttal to Reviewer Reports – NCOMMS-22-02826-T/Haykal

We would like to thank the three Reviewers for considering our manuscript for publication. We are encouraged by the positive comments of Reviewer #1 and Reviewer #3 who both suggest that publication can be considered after some issues are addressed. Reviewer #2 does not recommend publication, pointing out some inconsistencies between experimental results. We will explain below in details why we strongly disagree with the statements made by Reviewer #2.

In the following, we include a point-by-point response to all the questions and criticisms raised by the three Reviewers. Their remarks are included verbatim and typeset in *blue italics* for the sake of readability. Changes to the manuscript are highlighted in red in the resubmitted files.

We are confident we have addressed all pertinent remarks and clarified all issues in full. We therefore resubmit our revised manuscript for further consideration in the journal.

Reviewer #1

The Authors report on a joint experimental-theoretical research effort to probe the decoherence of boron monovacancy defects in hBN. hBN has been the subject of intense research efforts to identify and utilize quantum defects embedded in the material. In this work, the Authors first utilize monoisotopic hBN to strengthen the attribution of the broad 800 nm optical signature that exhibits ODMR to the boron monovacancy. Their argument is based on the hyperfine coupling to the neighboring nitrogen spins. To my pleasure, the Authors also addressed the superhyperfine coupling, which supports their argument. The Authors then study the spin coherence properties of the boron monovacancy and extract two different T₂ times in the different monoisotopic crystals. They attribute the difference in the T₂ times to the differing isotopes and demonstrate the importance of including the Fermi contact term in the CCE calculation, correcting a misconception in the literature. Lastly, the Authors probe additional sources of coherence, "dark spins", which introduce additional decoherence beyond the nuclear bath..

I find the work to be of high quality and fairly well done. A number of issues are addressed, and I was happy to see that the concerns I had while reading were mostly addressed soon after in the manuscript.

We thank the Reviewer for these positive comments on our work.

I just have one concern that should be addressed prior to publication because it is essential to the message of the manuscript.

The Authors give the impression that the T₂ times are extracted from a single defect in each of the two different samples that differ in isotopic purity. The difference in these T₂ times is attributed to the differing isotopes. However, it is premature to make such a conclusion if only two defects have been investigated. Differing local environments could instead lead to the differing T₂ times. Indeed, the Authors point out that other experiments have measured a range of T₂ values. To make an attribution to the isotope effect, the Authors should investigate a statistically significant number of defects to rule out the differing local environments and isolate the isotope effect. If this was already done, the Authors need to make this clear.

All experiments reported in our work were performed on large ensembles of V_B⁻ defects. It can be seen in Figure 1 (b) that the PL map is homogeneous all over the hBN sample, such that individual V_B⁻ defects cannot be isolated. More generally, the possibility to isolate individual V_B⁻ defects remains an open question owing to the very low quantum efficiency of the optical transition. To date all experiments performed on V_B⁻ defects have been realized on large ensembles. The T₂ measurements reported in our work therefore correspond to an average over a large number of spin defects. In addition, experiments performed at different locations of the isotopically-purified hBN crystals led to identical results. As suggested by the Reviewer, we made this clearer in the new version of the manuscript.

Reviewer #2

The authors of this work reported their creation and study of boron-vacancy spin defects in isotopically enriched hexagonal boron nitride (hBN). Spin defects in hBN provide a new paradigm of quantum systems (in van der Waals materials) that will have applications in quantum sensing and other fields. Previous research has shown that isotopic enrichment can increase the coherence of diamond nitrogen-vacancy centers. Thus we would hope isotopic enrichment can increase the coherence time of spin defects in hBN or add new functionalities. In this work, the authors used hBN crystals enriched with either boron-10 or boron-11 isotopes to study isotope-dependent properties of spin defects. By comparing the hyperfine structures with different boron isotopes, the authors confirmed that the created spin defects are negatively-charged boron vacancies as claimed by other works before. However, the coherence time of hBN spin defects reported in this work is much shorter than several former works and does not show the advantages of isotopic enrichment.

The authors attributed the discrepancy between their result of short electron spin coherence time (T_2) and several former results of much longer coherence times to possible mistakes made by other groups.

As discussed in our manuscript, only few papers in the literature have studied the spin coherence properties of V_B^- defects in hBN, leading to very different results. Some of these studies have indeed reported microsecond long coherence time. We have not written that the authors of these works have made “mistakes”. Given the novelty of the field, our intention is not to be controversial. We have solely indicated in our paper that the contrast of the echo signal was very weak in these experiments and that the experimental data at short time scale were not shown. In addition, such a long electron spin coherence time is not captured by our theoretical calculations, which carefully consider the Fermi contact term and first-order correlation effects. We suggest in our work that the decay of the weakly contrasted spin echo signal observed at long timescale by other groups could be related to coherences imprinted in nearby nuclear spins. This is an interesting point that will require further analysis going beyond the scope of the present work.

However, I found Fig. 3(c) was inconsistent with Fig. 3(a) in Figure 3 (Spin coherence properties) of this work. Fig. 3(a) in this work shows clear Rabi oscillations for times longer than 200 ns. Based on Fig. 3(a), the inhomogeneous coherence time (T_2^) is about 100 ns, consistent with former results. T_2 must be longer than T_2^* . Thus the coherence time T_2 should be much longer than 100 ns. However, Fig. 3(c) shows the T_2 to be only about 46 ns or 62 ns, much shorter than the T_2^* shown in Fig. 3(a). The inconsistency between Fig. 3(a) and Fig. 3(c) may be because the fast decay shown in Fig. 3(c) is not due to spin decoherence. For example, the laser and MW pulses might have finite rise and decay times and overlaps that affect the measurement. In addition, to measure T_2 , the MW pulse durations should be much shorter than T_2 . If the MW pulse durations were not short enough, the spins might have significant decoherence within each pulse.*

Fitting the envelope of the Rabi oscillation with an exponential function leads to a characteristic decay time $T_R \sim 100$ ns. We do not agree with the Reviewer that such a decay time corresponds to the coherence time T_2^* of the spin defect. Indeed, Rabi oscillation measurements mostly probe populations of the quantum system rather than its coherence, the latter being commonly inferred through Ramsey spectroscopy. Many examples can be found in the literature where the decay of the Rabi oscillation is much longer than T_2^* . For instance, Chiorescu *et al.* [Science **299**, 1869 (2003)] measure a Rabi decay time up to 150 ns for a superconducting flux qubit, while the T_2^* time obtained through Ramsey spectroscopy is around 20 ns and can be solely extended to 30 ns using spin echo techniques. For NV defects in diamond, Rabi oscillations are commonly obtained over several tens of microseconds while T_2^* does not exceed few microseconds [see for example Fedder *et al.* Appl. Phys. B **102**, 497 (2011)]. We therefore disagree that there is an inconsistency between the results shown in Fig. 3(a) and Fig. 3(c) of our paper. Following the comment of the Reviewer, we have added few sentences in the new version of the manuscript to stress that the decay of the Rabi oscillation cannot be used to infer the spin coherence time T_2^* .

The Reviewer also indicates that the observed decay of the spin-echo signal could result from experimental artefacts linked to the rising time of laser and microwave pulses. If such artefacts would be at play, they should also be observed in the Rabi measurements, which are recorded on similar timescales. In addition, potential experimental artefacts were excluded by performing coherent manipulation of a single NV defect in the same experimental setup.

We agree with the Reviewer that the microwave pulse duration must be shorter than the coherence time in spin-echo experiments. In our experiment, the duration of the $\pi/2$ pulse is set to 15 ns which is short enough to obtain reliable T_2 measurements. This information has been added in the new version of the paper.

The paper "Defect Engineering of Monoisotopic Hexagonal Boron Nitride Crystals via Neutron Transmutation Doping" [Chem. Mater. 2021, 33, 9231–9239] has reported ODMR with isotopically enriched hBN and should be cited in this work.

This reference has been added in the new version of the manuscript.

In summary, I think some conclusions of this work are not supported by its results, especially the inconsistency between Fig. 3(a) and Fig. 3(c). Isotopic enrichment may become important for hBN spin defects in the future. However, the current work has not shown clear advantages of isotopic enrichment. Thus I cannot recommend the publication of this work in Nature Communications.

As discussed in details above, we disagree that there is an inconsistency in the experimental results shown in Figure 3. As indicated by the Reviewer, we indeed show that isotopic purification does not lead to a dramatic improvement of the spin coherence properties of spin defects in hBN. This is one of the main messages of our work, which is supported both by spin-echo measurements and theoretical calculations. We believe that this is a very important information for future developments of spin defects in hBN for quantum technologies. Indeed, there was a hope of improving spin coherence properties by isotopic purification similar to what can be achieved in diamond. However, isotopic purification in diamond is radically different since it is done with ^{12}C atoms (spinless). Being nestled in a nuclear spin-free diamond lattice, the coherence time of spin defects are significantly improved. In hBN the situation is different since each lattice site is occupied by an atom with non-zero nuclear spin. Isotopic purification with either ^{10}B or ^{11}B only change the nuclear spin gyromagnetic ratio and modifies the dynamics of the nuclear spin bath. Our work shows that it leads only to a slight improvement of coherence properties. This is supported by numerical simulations employing state-of-the-art cluster correlation expansion methods. These simulations highlight the importance of the spin density distribution, the corresponding Fermi-contact hyperfine fields, and first order correlation effect for determining the coherence time of a solid-state spin qubit interacting with a dense nuclear spin bath. Beyond the V_{B}^- center in hBN, this finding generally applies to any defect hosted in a solid with a high nuclear spin density.

Reviewer #3

This manuscript from A. Haykal et al. reports on the experimental measurements of the spin coherence in isotopically enriched hBN samples. This work studies how the coherence properties of the V_{B}^- defect in hBN change when the material contains ^{10}B or ^{11}B . The authors interface experimental and theory to identify the spin structure of the defects through the analysis of the hyperfine structure. Overall, this paper is well written, the results are interesting and presented clearly. I think the topic is suitable for publication in Nature Communications.

We thank the Reviewer for these positive assessments on our work.

However, I have one major comment that should be addressed before I can recommend publication. The authors claim to unambiguously confirm that the investigated defects, created by thermal neutron irradiation, correspond to the V_{B}^- . This claim is based on the observation of 7 hyperfine lines in the ODMR spectrum. As the authors state at pag.8, this can be explained either by a B vacancy or a substitutional impurity with $I=0$. It seems that the conclusion about the observation of B vacancy is based on the fact that neutron transmutational doping with irradiation "likely produces boron vacancy-related centers". Although the creation of B vacancies is likely, this does not seem to exclude the possibility to have also created substitutional defects with $I=0$. At pag. 15, the authors comment on how neutron irradiation can create other defects centers with $S=1/2$, including defects with C inclusion. I think this point needs to be clarified in the paper. In order to "unambiguously" claim the V_{B}^- defects, I would suggest to discuss why no other substitutional defects with $I=0$ could be created during the neutron irradiation.

As discussed in our paper, the observation of similar hyperfine spectra while changing the boron isotope indicates that the spin defect is either a V_{B}^- center or a substitutional impurity with zero nuclear spin localized at a boron site. We agree with the Reviewer that it is difficult to discriminate between these two possibilities.

Substitutional carbon C_B could be a plausible candidate, since carbon is a well-identified contaminant in hBN with a dominant spinless isotope (^{12}C , 99%). The formation of such defects could be obtained through the creation of boron vacancies via neutron irradiation followed by the migration of substitutional carbon impurities at boron sites. This process seems unlikely given that our monoisotopic hBN crystals were not annealed after irradiation thus preventing any migration of impurities. In addition, *ab-initio* theory of the V_B^- center reproduces very well all our experimental results, including a broad PL emission in the near infrared, a zero-field splitting parameter around $D \sim 3.5$ GHz, and a hyperfine coupling of 47 MHz with the first neighbor ^{14}N nuclei. Furthermore, the hyperfine spectra simulated in Figure 2(d,e) while including the hyperfine coupling constants calculated up to the third neighbors reproduce fairly well our experimental data. On the other hand, C_B defects have a very different hyperfine signature, as discussed in the paper by Ph. Auburger *et al.*, Phys. Rev. B **104**, 075410 (2021). We thus concluded that the defect corresponds to the V_B^- center. To take into account the reviewer's comment, we have added this discussion and we have removed the word '*unambiguously*' in the new version of the manuscript.

REVIEWER COMMENTS

Reviewer #1 (Remarks to the Author):

The Authors have addressed my concerns by clarifying that their measurement of the T2 time is based on ensembles and have edited their manuscript accordingly. I think the work is of good quality and warrants publication in Nature Communications.

Reviewer #2 (Remarks to the Author):

This work does contain some valuable studies about spin defects in monoisotopic hexagonal boron nitride. However, the authors' explanation about the inconsistency between T2* inferred from Rabi oscillation in Fig. 3(a) and T2 in Fig. 3(c) is not satisfactory. I agree with the authors that Ramsey spectroscopy will be better for determining T2* than the Rabi oscillation. As this will be important for clarifying the discrepancy, the authors should add results of Ramsey spectroscopy, which should be easy to do.

There has been an arXiv paper ("Rabi oscillation of VB- spin in hexagonal boron nitride", arXiv:2101.11220) that reported measurements of the Ramsey interference of boron vacancy spin defects in hBN. With Ramsey interference, it reported a T2* of 60ns at 0 mT, and around 1 microsecond at 44 mT, which is longer than T2 reported in this paper. Considering the discrepancy, it will be important for the authors to perform Ramsey interference under the same conditions as spin echos in Fig. 3(c).

Reviewer #3 (Remarks to the Author):

In the revised manuscript, the authors have addressed my previous comments and I can now recommend publication.

Authors' Rebuttal to Reviewer Reports – NCOMMS-22-02826A/Haykal

We would like to thank the three Reviewers for considering our manuscript for publication. While Reviewer #1 and Reviewer #3 both support publication of our work, Reviewer #2 have one additional comment which we respond to below.

The Reviewers remarks are included verbatim and typeset in *blue italics* for the sake of readability. Changes to the manuscript are highlighted **in red** in the resubmitted files.

We are confident we have addressed the pertinent remarks of Reviewer #2 and clarified all issues in full. We therefore resubmit our revised manuscript for further consideration in the journal.

Reviewer #1

The Authors have addressed my concerns by clarifying that their measurement of the T2 time is based on ensembles and have edited their manuscript accordingly. I think the work is of good quality and warrants publication in Nature Communications.

We thank the Reviewer for these positive comments.

Reviewer #2

This work does contain some valuable studies about spin defects in monoisotopic hexagonal boron nitride. However, the authors' explanation about the inconsistency between T2 inferred from Rabi oscillation in Fig. 3(a) and T2 in Fig. 3(c) is not satisfactory. I agree with the authors that Ramsey spectroscopy will be better for determining T2* than the Rabi oscillation. As this will be important for clarifying the discrepancy, the authors should add results of Ramsey spectroscopy, which should be easy to do.*

There has been an arXiv paper ("Rabi oscillation of VB- spin in hexagonal boron nitride", arXiv:2101.11220) that reported measurements of the Ramsey interference of boron vacancy spin defects in hBN. With Ramsey interference, it reported a T2 of 60ns at 0 mT, and around 1 microsecond at 44 mT, which is longer than T2 reported in this paper. Considering the discrepancy, it will be important for the authors to perform Ramsey interference under the same conditions as spin echos in Fig. 3(c).*

As suggested by the Reviewer, we have performed Ramsey spectroscopy under the same experimental conditions used to record the spin echo data shown in Fig. 3(c). A typical free induction decay has been added as a supplementary Figure. This measurement indicates a spin dephasing time $T_2^* \sim 20$ ns, which is much shorter than the decay of the Rabi oscillation. We hope that this additional measurement will convince the reviewer that there are no inconsistencies in our experimental results.

Reviewer #3

In the revised manuscript, the authors have addressed my previous comments and I can now recommend publication.

We thank the Reviewer for this positive assessment on our work.

REVIEWERS' COMMENTS

Reviewer #2 (Remarks to the Author):

The authors had performed Ramsey spectroscopy and included the result in the supplementary file (supplementary Figure 1) as I suggested. I can now recommend publication.